# Efficacy of Dialkylcarbamoylchloride (DACC)-Impregnated Dressings in Surgical Wound Management: A Review

**DOI:** 10.3390/ebj6010001

**Published:** 2025-01-21

**Authors:** Madhan Jeyaraman, Naveen Jeyaraman, Swaminathan Ramasubramanian, Arulkumar Nallakumarasamy, Shrideavi Murugan, Tarun Jayakumar, Sathish Muthu

**Affiliations:** 1Department of Orthopaedics, ACS Medical College and Hospital, Dr MGR Educational and Research Institute, Chennai 600077, Tamil Nadu, India; madhanjeyaraman@gmail.com; 2Orthopaedic Research Group, Department of Orthopaedics, Coimbatore 641045, Tamil Nadu, India; 3Department of Orthopedics, Brazilian Institute of Regenerative Medicine (BIRM), Indaiatuba 13334-170, SP, Brazil; 4Department of Orthopaedics, Government Medical College, Omandurar Government Estate, Chennai 600002, Tamil Nadu, India; swaminathan.ramasubramanian@outlook.com; 5Department of Orthopaedics, Jawaharlal Institute of Postgraduate Medical Education and Research (JIPMER), Karaikal 609602, Puducherry, India; arulmmcian@gmail.com; 6Department of Orthopaedics, Government Tirunelveli Medical College and Hospital, Tirunelveli 627002, Tamil Nadu, India; shrideavimurugan@gmail.com; 7Department of Orthopaedics, KIMS-Sunshine Hospital, Hyderabad 500032, Telangana, India; tarunjaykumar@gmail.com; 8Department of Orthopaedics, Government Medical College and Hospital, Karur 639004, Tamil Nadu, India; 9Department of Biotechnology, Faculty of Engineering, Karpagam Academy of Higher Education, Coimbatore 641021, Tamil Nadu, India

**Keywords:** dialkylcarbamoyl chloride, surgical site infections, wound healing, antimicrobial resistance, wound dressings, cost-effectiveness

## Abstract

Surgical site infections (SSIs) are a significant challenge in postoperative care, leading to increased morbidity, extended hospital stays, and elevated healthcare costs. Traditional antimicrobial dressings, such as those containing silver or iodine, have limitations, including cytotoxicity and the potential for antimicrobial resistance. Dialkylcarbamoyl chloride (DACC)-impregnated dressings offer a novel approach, employing a physical mechanism to bind and remove bacteria without the use of chemical agents, thereby reducing the risk of resistance. This review summarizes current evidence on the efficacy of DACC dressings in preventing SSIs and promoting wound healing. Findings from multiple studies indicate that DACC dressings reduce bacterial burden and SSI rates across various surgical procedures, including cesarean sections and vascular surgeries. Additionally, DACC dressings demonstrate potential in managing hard-to-heal wounds, such as diabetic foot ulcers, by reducing bacterial load and biofilm formation. Furthermore, they present advantages in antimicrobial stewardship and cost-effectiveness by minimizing the need for antibiotics and decreasing overall healthcare expenses. However, the current literature is limited by small sample sizes, methodological weaknesses, heterogeneity in study designs, and a lack of long-term data. Future research should focus on high-quality randomized controlled trials across diverse surgical populations, comprehensive cost-effectiveness analyses, and long-term outcomes to establish the full clinical impact of DACC dressings. With further validation, DACC-impregnated dressings could become a critical tool in sustainable postoperative wound care.

## 1. Introduction

Surgical site infections (SSIs) are a critical concern in postoperative care, significantly impacting patient outcomes and healthcare systems. SSIs account for 5–20% of hospital-acquired infections, leading to increased morbidity, extended hospital stays, higher readmission rates, and elevated healthcare costs [1]. The management of surgical wounds is vital in preventing these complications. Traditional wound care methods, such as gauze dressings, aim to absorb exudate and protect the wound from contamination, yet they are often insufficient in preventing infections, particularly in high-risk surgeries. Antimicrobial dressings, including those impregnated with silver or iodine, have been introduced to reduce microbial load in wounds [2]. However, these substances carry limitations. Silver, for instance, can cause cytotoxicity and delay wound healing, while iodine can be harsh on tissue, especially in chronic wounds [3,4]. Moreover, these dressings rely on chemical agents that may contribute to antimicrobial resistance over time [5]. Dialkylcarbamoyl chloride (DACC)-coated dressings have emerged as an alternative, utilizing a physical mechanism to bind and remove bacteria without contributing to resistance or releasing harmful endotoxins [6]. Despite their promise, high-quality, large-scale studies are needed to fully validate their efficacy and cost-effectiveness in wound management across diverse patient populations.

DACC-impregnated dressings represent a novel approach to wound care by utilizing a purely physical mechanism to prevent microbial colonization. DACC’s mechanism of action is based on hydrophobic interactions, which exploit the inherent hydrophobicity of bacterial cell walls [7]. Most pathogenic bacteria, including Gram-positive and Gram-negative strains, possess hydrophobic surfaces that naturally repel water. DACC-coated fibers in the dressings create a hydrophobic environment that attracts and irreversibly binds these bacteria. Once bound to the DACC-coated dressing, bacteria are physically trapped and removed from the wound site when the dressing is changed [6]. This interaction prevents bacteria from adhering to the wound bed, thereby inhibiting biofilm formation and subsequent infection [7,8]. Unlike traditional antimicrobial dressings that rely on chemical agents like silver or iodine, DACC does not release any active substances into the wound [9,10]. This means it does not contribute to antimicrobial resistance, a major concern with chemical-based treatments. Because DACC-coated dressings do not release bactericidal agents, they avoid cytotoxic effects on host cells, allowing normal wound healing processes to occur without disruption [6,7]. By mechanically reducing microbial load, DACC dressings offer a safer and more sustainable option in the prevention and management of wound infections.

A review is necessary to summarize the expanding body of evidence on DACC-impregnated dressings in surgery, as their unique mechanism holds significant potential to improve infection control in postoperative care. Despite promising findings, research on DACC dressings remains fragmented across various studies, making it essential to systematically map the available data. This review will help determine the breadth of evidence, identify gaps, and assess the clinical impact of DACC dressings, guiding future research and influencing surgical wound management practices. The objective is to map the existing evidence on the efficacy of DACC-impregnated dressings in preventing surgical site infections and improving wound healing in surgical settings.

## 2. Methodology

To conduct a comprehensive review on DACC-coated dressings, we searched for relevant articles in PubMed, Embase, Web of Science, and Scopus databases. We utilized specific search terms such as “DACC”, “Dialkylcarbamoyl chloride”, “silver-coated”, “iodine”, “honey”, “dressings”, and “effectiveness”. These terms were combined using Boolean operators (AND, OR, NOT) to refine our search results, and we applied filters to narrow down the results by publication date, article type, and language. We carefully reviewed the titles and abstracts of the search results to identify pertinent articles, then obtained and reviewed the full-text articles for in-depth information. Additionally, we examined the reference lists of selected articles to locate further relevant sources. By thoroughly examining all the four databases, we ensured a comprehensive collection of source articles for our review.

## 3. DACC-Coated Dressings

DACC-coated dressings have emerged as a promising tool in wound management, primarily due to their unique mechanism of action, which binds and removes bacteria from the wound bed without releasing antimicrobial agents [6]. This characteristic positions DACC dressings as a potential alternative to traditional dressings, especially in the context of antimicrobial resistance (AMR) and the limitations of conventional wound care strategies, such as those involving silver or iodine dressings.

### 3.1. General Efficacy of DACC-Coated Dressings

Across multiple studies, DACC-coated dressings have shown significant efficacy in reducing bacterial burden in wounds, particularly against pathogens prioritized by the World Health Organization (WHO), such as *Staphylococcus aureus* and *Pseudomonas aeruginosa* [9]. The dressings work through hydrophobic interactions, binding bacteria to the dressing surface without disrupting their cell walls. This prevents the release of bacterial endotoxins, a key contributor to inflammation and delayed healing in infected wounds [6]. Unlike traditional antimicrobial dressings, DACC-coated dressings do not rely on chemical agents to kill bacteria, which reduces the risk of developing antimicrobial resistance—a major advantage in the current healthcare landscape (Figure 1) [11]. The application of DACC-coated dressings is depicted in Figure 2.

### 3.2. Clinical Outcomes in Surgical Site Infections

A significant body of evidence supports the use of DACC dressings in reducing SSIs across different types of surgeries. For instance, a randomized controlled trial (RCT) involving 543 women undergoing cesarean sections demonstrated that the application of DACC dressings reduced the SSI rate to 1.8%, compared to 5.2% in the group receiving standard surgical dressings (SSD) (*p* = 0.04). Notably, the cost of treating SSIs was also significantly lower in the DACC group (1065 EUR) compared to the SSD group (5775 EUR) [12]. Similar findings were observed in studies of vascular surgery, where DACC-coated dressings were associated with a reduction in early SSIs, with an infection rate of 1% at five days compared to 10% in the control group (*p* < 0.05) [13]. Although the RCT evidence is limited to the above studies, a recent summary of retrospective evidence also indicates that DACC dressings perform particularly well in preventing early post-operative infections, reducing both morbidity and associated healthcare costs [6,9,11].

### 3.3. Performance in Hard-to-Heal Wounds

DACC-coated dressings have shown promise in treating difficult-to-heal lesions such as diabetic foot ulcers and chronic leg ulcers [14,15,16]. A systematic review of 17 studies, including RCTs and case series, highlighted the significant reduction in bacterial load in chronic wounds treated with DACC dressings compared to silver-coated dressings [6]. One pilot RCT in particular reported a 73.1% reduction in bacterial load in chronic leg ulcers treated with DACC dressings, compared to 41.6% in wounds treated with silver dressings (*p* < 0.01) [17]. This bacteriostatic effect, coupled with the ability to bind biofilms and remove bacteria without promoting resistance, underscores the utility of DACC dressings in managing chronic wounds [7,14,18]. However, the review also noted that more high-quality studies are needed to fully establish their clinical and cost-effectiveness. The summary of findings from the studies included in the review is presented as Table 1.

### 3.4. Effectiveness in Pilonidal Sinus Disease

The performance of DACC dressings in promoting wound healing has also been evaluated in pilonidal sinus disease. In a multicenter RCT involving 246 patients, 75.7% of wounds treated with DACC dressings had completely healed after 75 days, compared to 60% in the group treated with alginate dressings (*p* = 0.023). Despite the statistically significant improvement in healing rates, the preplanned clinically significant improvement of 20% was not achieved. While patient-reported outcomes, including comfort and leakage, did not differ significantly between the two groups, the findings suggest that DACC dressings may offer advantages in terms of faster healing and reduced infection risk [34].

### 3.5. Antimicrobial Resistance and Stewardship

One of the primary advantages of DACC-coated dressings lies in their role in antimicrobial stewardship, an increasingly important consideration in the global fight against antimicrobial resistance (AMR). DACC dressings do not rely on chemical agents to combat infection; instead, they use a physical mechanism to bind bacteria through hydrophobic interactions, effectively removing them from the wound bed [7,10]. This method reduces the risk of bacterial colonization and infection without contributing to the development of resistant strains, a significant concern with traditional antimicrobial dressings that rely on chemical agents such as silver or iodine [35]. Silver-coated dressings, for instance, release silver ions that are toxic to bacterial cells, offering broad-spectrum antimicrobial effects [3,36,37,38]. However, the widespread and prolonged use of silver-based dressings raises concerns about the potential for bacterial resistance. Studies have shown that bacteria exposed to sub-lethal concentrations of silver may develop resistance over time, diminishing the long-term efficacy of silver-based antimicrobial agents [39,40,41,42]. This issue is of growing concern as healthcare systems grapple with rising levels of drug-resistant pathogens. In contrast, DACC dressings, by maintaining bacterial integrity while physically binding and removing them from the wound, offer a solution that minimizes the risk of resistance [13]. This aligns with global health initiatives, such as those advocated by the World Health Organization (WHO), which emphasize the reduction of antimicrobial use to combat AMR and promote sustainable infection management strategies. Iodine-based dressings, another commonly used antimicrobial option, similarly rely on chemical agents to eradicate bacteria. While iodine is effective against a broad range of pathogens, including bacteria, viruses, and fungi, its prolonged use has been associated with the potential for selecting resistant strains [4,43]. Moreover, iodine can cause irritation and hypersensitivity reactions in some patients, limiting its use in individuals with allergies or sensitive skin [44,45,46,47,48]. DACC dressings, by avoiding chemical agents altogether, offer a safer alternative that significantly reduces the likelihood of developing resistant microorganisms while also minimizing adverse reactions in patients.

### 3.6. Early Post-Operative Infection Prevention

DACC dressings have been shown to outperform standard dressings in reducing the risk of surgical site infections (SSIs) during the critical early post-operative period, a time when wounds are particularly vulnerable to bacterial colonization [13]. Several studies have highlighted the effectiveness of DACC dressings in lowering infection rates, particularly in high-risk surgeries such as cesarean sections and vascular surgery. A study on cesarean sections indicated that patients treated with DACC dressings had a considerably reduced rate of SSIs than those treated with normal surgical dressings. In this study, the DACC group demonstrated an infection rate of 1.8%, while the standard dressing group had a significantly higher infection rate of 5.2% (*p* = 0.04) [12]. This reduction in infection rates is particularly important in the first few days following surgery, when the risk of bacterial colonization is highest. Standard surgical dressings, typically made of absorbent materials like cotton, provide minimal protection against bacterial infiltration, making them less effective at preventing early infections [49]. In contrast, DACC dressings, through their unique ability to bind and remove bacteria, offer enhanced protection against infections during this critical period [17]. Similarly, in vascular surgeries, where the risk of SSIs is elevated due to the presence of prosthetic materials and prolonged surgical exposure, DACC dressings have demonstrated superior infection control compared to alginate and standard surgical dressings [10]. Alginate dressings, while effective in managing wound exudate, lack antimicrobial properties and are therefore less effective at preventing bacterial colonization in the wound bed [50]. DACC dressings, by offering both moisture management and infection control, provide a more comprehensive solution for post-operative wound care.

### 3.7. Cost-Effectiveness

Another key advantage of DACC dressings is their cost-effectiveness, particularly in the context of surgical and chronic wound management. By reducing infection rates and the associated need for additional treatments, including antibiotics, DACC dressings help alleviate the financial burden on healthcare systems. Several studies have highlighted the economic benefits of DACC dressings, particularly in high-risk surgical populations. In a study on cesarean sections, the total SSI-related costs were significantly lower in the DACC group compared to the standard surgical dressing group. This reduction in costs was attributed to the lower infection rates observed in the DACC group, which resulted in fewer hospital readmissions, reduced antibiotic use, and shorter durations of wound management [12]. While DACC dressings may have a higher initial cost compared to standard dressings, the long-term savings associated with reduced infection rates and the need for fewer medical interventions make them a cost-effective option for healthcare providers [35,51]. Similarly, in chronic wound management, where patients often require long-term care and multiple dressing changes, DACC dressings have been shown to reduce the overall cost of treatment by promoting faster wound healing and minimizing the need for antibiotics [6]. Chronic wounds, such as diabetic foot ulcers and pressure ulcers, are particularly susceptible to bacterial colonization and infection, which can delay healing and increase the cost of care. By effectively binding and removing bacteria, DACC dressings help create a more favorable environment for wound healing, reducing the need for prolonged treatment and associated healthcare costs [9,52,53,54]. The advantages of DACC-coated dressings are depicted in Figure 3.

## 4. Comparison with Antimicrobial Dressings

DACC dressings offer several advantages over traditional antimicrobial dressings, including silver-coated, iodine-based, and honey-based dressings, particularly in terms of antimicrobial resistance, cytotoxicity, and wound healing as given in Table 2.

### 4.1. Silver-Coated Dressings

As previously mentioned, silver-coated dressings release silver ions, which are toxic to bacterial cells and help reduce infection risk. However, silver dressings are associated with several drawbacks, including the potential for inducing bacterial resistance, cytotoxicity, and higher costs [55]. One of the main concerns with silver-coated dressings is their potential to contribute to antimicrobial resistance. Prolonged use of silver dressings has been shown to select for resistant bacterial strains, reducing their long-term effectiveness in preventing infections [56]. DACC dressings, by contrast, offer a non-chemical mechanism of action that does not contribute to resistance, making them a safer option for long-term wound management [51].

In addition to concerns about resistance, silver ions can also be cytotoxic, meaning they can damage human cells, including fibroblasts and keratinocytes, which are essential for wound healing [57]. This cytotoxicity can delay wound closure and prolong healing, particularly in chronic wounds. DACC dressings, by maintaining bacterial integrity while physically removing them from the wound, promote a more favorable healing environment without damaging surrounding tissue [35]. Silver dressings tend to be more expensive than DACC dressings, making them a less attractive option for routine wound care, particularly in cases where long-term management is required. The higher cost, combined with the potential for cytotoxicity and resistance, makes DACC dressings a more cost-effective and safer alternative for surgical and chronic wound care [30,58,59,60].

### 4.2. Iodine-Based Dressings

Iodine-based dressings, such as povidone-iodine, offer broad-spectrum antimicrobial effects and are commonly used in wound care [61]. However, like silver dressings, iodine dressings rely on chemical agents to kill pathogens, which can lead to the development of resistant strains with prolonged use [62,63]. Moreover, iodine can cause irritation and hypersensitivity reactions in some patients, limiting its use in individuals with allergies or sensitive skin [44]. These adverse reactions can lead to discomfort, delayed wound healing, or even complications that require additional medical intervention. DACC dressings, by avoiding chemical agents, offer a safer alternative for patients with sensitive skin or allergies and are not associated with such adverse effects [13]. In terms of wound healing, iodine-based dressings may negatively affect the wound healing process if used over extended periods [64]. DACC dressings, by reducing bacterial bioburden without disrupting the healing process, offer a more suitable option for long-term wound management, particularly in surgical wounds that require extended healing periods [65,66].

### 4.3. Honey-Based Dressings

Honey, particularly medical-grade Manuka honey, is another antimicrobial agent used in wound care. Honey dressings create an acidic environment and generate hydrogen peroxide, which inhibits bacterial growth [67,68]. However, the antimicrobial effectiveness of honey dressings can be variable depending on the type and concentration of honey used, making them less predictable than DACC dressings [69]. While honey has been shown to promote wound healing by maintaining a moist environment and supporting autolytic debridement, honey dressings can be sticky and uncomfortable during dressing changes, especially in sensitive surgical wounds [70]. DACC dressings, by contrast, are easy to apply and remove without causing trauma to the wound bed, making them a more patient-friendly option for long-term use [52].

## 5. Comparison with Non-Antimicrobial Dressings

DACC dressings also outperform non-antimicrobial dressings, such as alginate and standard surgical dressings, particularly in terms of infection control and wound healing [34].

### 5.1. Alginate Dressings

Alginate dressings, derived from seaweed, are frequently used in wound care due to their excellent absorbent properties, particularly in heavily exudating wounds. However, alginate dressings do not offer antimicrobial properties, making them less effective in preventing infections in surgical wounds where infection risk is high [51]. DACC dressings, by providing both moisture management and infection control, offer a more comprehensive solution for wound care, particularly in surgical wounds that require both exudate absorption and antimicrobial action. Additionally, studies have shown that DACC dressings promote faster wound healing compared to alginate dressings, particularly in high-risk surgical scenarios such as pilonidal sinus excision [34,71].

### 5.2. Standard Surgical Dressings

Standard surgical dressings (SSD) are commonly used to cover and protect wounds post-operatively [72]. However, these dressings, typically made of cotton or other absorbent materials, do not offer antimicrobial properties, making them less effective at preventing SSIs compared to DACC dressings [73]. In studies comparing DACC dressings with standard surgical dressings, DACC dressings have consistently demonstrated superior infection control. For example, in women undergoing cesarean sections, the SSI rate was significantly lower in the DACC group compared to the SSD group, highlighting the effectiveness of DACC dressings in preventing post-operative infections [12].

## 6. Gaps in Literature

Despite the promising outcomes associated with DACC-coated dressings, there are considerable gaps in the existing literature that impede a full understanding of their long-term efficacy and broader applicability in wound management. These gaps span across long-term follow-up, evaluation in specific high-risk surgical populations, comprehensive cost-effectiveness assessments, and the role of DACC in mitigating AMR.

### 6.1. Long-Term Follow-Up and Outcomes

One of the most pressing gaps in the current literature is the limited availability of long-term data on the effectiveness of DACC-coated dressings. Most studies focus on short-term outcomes, such as wound healing rates or infection control within the initial 30 to 75 days post-application [51]. While these findings are valuable for immediate clinical decision making, they fall short of providing a complete picture of the long-term benefits and challenges associated with DACC dressings. The absence of long-term follow-up is particularly concerning for chronic wounds, such as diabetic foot ulcers or venous leg ulcers, which often require extended treatment periods and are prone to recurrent infections [17,29,51]. The high recurrence rate of infection in chronic wound patients, coupled with the tendency for these wounds to persist over months or even years, necessitates data that extend well beyond the typical study periods currently reported. Longitudinal studies with follow-up periods extending to six months, a year, or even longer would provide critical insights into the durability of DACC dressing effects, such as infection recurrence, wound closure sustainability, and prevention of long-term complications like scarring or non-healing wounds. This gap is also evident in studies involving acute wounds, such as those in surgical patients. Research predominantly focuses on immediate post-operative periods, often up to 30 days, with scant attention to outcomes beyond this window [74]. The limited data on long-term infection prevention and wound healing in surgical patients means that the potential benefits of DACC dressings in reducing complications like late-onset surgical site infections (SSIs) or preventing scar tissue formation remain underexplored.

### 6.2. Specific Surgical Populations

Another notable gap in the literature is the lack of research evaluating the efficacy of DACC dressings in a broader range of surgical populations. Most of the available evidence focuses on relatively limited groups, such as patients undergoing cesarean sections, vascular surgery, and treatment for pilonidal sinus disease [12,13,34]. While these studies provide foundational evidence for the utility of DACC dressings in reducing SSIs, they do not necessarily translate to other surgical fields. For instance, high-risk surgical groups, such as those undergoing orthopedic, cardiac, or gastrointestinal surgeries, remain largely unexplored in the context of DACC dressings. These populations are particularly vulnerable to infections due to factors such as prolonged operative times, the use of implants, or compromised immune function [75,76]. For example, orthopedic surgeries, especially those involving joint replacements, present a significant infection risk due to biofilm formation around implants [77]. Given that DACC dressings have shown potential in preventing biofilm-associated infections, research in this area could clarify their potential for improving post-operative outcomes in orthopedic surgery patients [8,78]. Similarly, immunocompromised patients, such as those undergoing cancer treatments or organ transplants, are at heightened risk for infection, yet there is a lack of robust studies on the use of DACC dressings in these populations. Investigating the use of DACC dressings in immunocompromised groups could provide valuable data on their ability to prevent infections in individuals with reduced immune defenses. Further, the potential of DACC dressings to reduce bacterial colonization without the use of chemical agents might be particularly advantageous for these patients, who may be at greater risk of adverse effects from antimicrobial agents.

### 6.3. Cost-Effectiveness Data

Although preliminary data suggest that DACC dressings may offer cost savings in wound care, particularly by reducing the incidence of SSIs and minimizing the need for antibiotics, the current literature does not provide a comprehensive economic evaluation of their long-term cost-effectiveness. Existing studies tend to focus on short-term financial benefits, such as lower immediate healthcare costs due to reduced infection rates, shorter hospital stays, or fewer readmissions. While these outcomes are encouraging, they do not provide a complete picture of the financial impact of using DACC dressings over the long term. Comprehensive cost-effectiveness studies should consider not only the immediate financial savings but also the broader economic implications of using DACC dressings. Factors such as quality of life improvements, prevention of recurrent infections, and reduced costs associated with managing chronic wounds need to be taken into account. For instance, in the management of diabetic foot ulcers or venous leg ulcers, where wounds often take months to heal and are prone to complications; long-term savings related to fewer wound complications; lower infection recurrence rates; and reduced need for surgical interventions could significantly impact the cost-effectiveness profile of DACC dressings [79]. A randomized controlled trial comparing DACC dressings to standard dressings found that DACC dressings reduced the SSI rate by 64% [32]. This reduction in SSIs translates to fewer hospital bed days, outpatient visits, and use of systemic antibiotics, resulting in overall cost savings. For instance, the model predicted potential cost savings of GBP 15.7 million per year and a reduction of approximately 9000 bed days for the NHS [32]. Additionally, studies need to evaluate the cost-effectiveness of DACC dressings in diverse healthcare settings, particularly in low-resource environments where the cost of wound care can be a critical factor in decision making. Current data on the cost-effectiveness of DACC dressings predominantly come from high-income countries [12,32,79]. Studies conducted in low- and middle-income countries, where wound care resources are limited, would provide valuable insights into the global applicability and financial viability of DACC dressings. Furthermore, head-to-head cost-effectiveness comparisons with other commonly used antimicrobial dressings, such as silver or iodine-based dressings, are needed to establish their relative economic benefits.

### 6.4. Impact on Antimicrobial Resistance (AMR)

While DACC dressings are frequently mentioned as tools for antimicrobial stewardship, the direct evidence linking their use to a reduction in AMR remains limited. Current studies primarily focus on the ability of DACC dressings to reduce bacterial load and prevent infections, but they do not adequately explore how these dressings influence the broader issue of AMR in the clinical environment. The hydrophobic nature of DACC dressings, which physically binds and removes bacteria from the wound site without the use of chemical antimicrobial agents, theoretically contributes to AMR mitigation by reducing the selective pressure on bacterial populations to develop resistance. However, few studies have specifically evaluated how the widespread use of DACC dressings might impact microbial resistance patterns over time. Future research should aim to directly assess the role of DACC dressings in preventing the emergence of antimicrobial-resistant organisms. Studies could, for example, track changes in microbial flora and resistance profiles in wounds treated with DACC dressings compared to those managed with traditional antimicrobial dressings. Additionally, research should investigate whether the routine use of DACC dressings in environments with high AMR risk, such as hospitals with known multidrug-resistant organism outbreaks, can lead to reductions in resistant infections.

### 6.5. Limitations of the Evidence

Although the evidence base supporting the use of DACC-coated dressings in wound care is growing, there are several important limitations that must be addressed. These include issues related to small sample sizes, methodological weaknesses, heterogeneity in study designs and outcomes, and potential biases in the existing studies.

#### 6.5.1. Small Sample Sizes

A significant limitation of many of the available studies on DACC dressings is the small sample sizes, which undermine the statistical power and generalizability of the findings. For example, studies on hard-to-heal wounds, such as diabetic foot ulcers, often include fewer than 100 participants, which limits the ability to draw firm conclusions about the efficacy of DACC dressings in this population. Small sample sizes increase the risk of Type I and Type II errors, potentially leading to an overestimation or underestimation of treatment effects. This limitation is especially pronounced in pilot studies, which are typically designed to assess the feasibility of larger trials rather than generate robust clinical evidence. While pilot trials are useful for identifying potential benefits and guiding future research directions, they cannot substitute for well-powered randomized controlled trials (RCTs). Without larger sample sizes, it is difficult to determine whether the positive outcomes observed in small studies would hold true across more diverse patient populations or in real-world clinical settings.

#### 6.5.2. Methodological Weaknesses

Several studies in the evidence base exhibit methodological weaknesses that limit the overall quality of the evidence. These weaknesses include non-randomized study designs, lack of blinding, and short follow-up periods, all of which contribute to potential bias and reduce the internal validity of the findings. Non-randomized studies, such as cohort studies or case series, are more susceptible to bias, including selection bias and confounding factors. For instance, some studies comparing DACC dressings to standard dressings for SSIs used historical controls or non-randomized allocation, which can introduce selection bias and make it difficult to isolate the effects of DACC dressings from other variables that may influence wound healing [53]. Additionally, the lack of randomization in some studies increases the likelihood that differences in patient characteristics, such as comorbidities or baseline infection risk, may skew the results. Blinding is another critical issue in many of the studies. In several trials, blinding of patients and clinicians was not feasible due to the distinct appearance of DACC-coated dressings compared to standard dressings. This lack of blinding introduces the possibility of observer bias, where knowledge of the dressing type could influence the assessment of wound healing outcomes. For example, clinicians may subconsciously rate wounds treated with DACC dressings more favorably if they expect the dressing to be effective. The inability to blind in these studies raises concerns about the reliability of the reported benefits. The short follow-up periods in many studies also limit the ability to assess the long-term effectiveness of DACC dressings. In some cases, follow-up periods extend only to 30 days post-operation or wound treatment, which may not be sufficient to capture important outcomes such as infection recurrence, complete wound healing, or long-term complications like scarring.

#### 6.5.3. Heterogeneity in Study Designs and Outcomes

Another limitation of the existing evidence is the significant heterogeneity across studies in terms of both study design and reported outcomes. Different studies use varying methods for assessing key outcomes such as wound healing, bacterial load reduction, and cost-effectiveness, which complicates efforts to synthesize the data and draw definitive conclusions about the relative effectiveness of DACC dressings. For example, some studies focus primarily on the reduction of bacterial bioburden as the primary outcome, while others emphasize clinical endpoints such as wound healing rates or healthcare costs. Additionally, the methods used to measure these outcomes vary widely across studies, with some using subjective clinical assessments of wound healing and others relying on quantitative measures such as bacterial colony counts. This lack of standardization makes it difficult to compare results across studies or to perform meta-analyses that could provide stronger evidence for the clinical effectiveness of DACC dressings. Moreover, the patient populations and wound types included in the studies are diverse, further contributing to the heterogeneity of the evidence. While some studies focus on specific wound types, such as SSIs in cesarean sections, others include a mix of acute and chronic wounds, each with different healing trajectories and infection risks. The differences in follow-up durations, ranging from 30 to 75 days, and sometimes longer, also introduce variability in reported outcomes.

#### 6.5.4. Potential Biases

Several studies exhibit potential biases that may affect the reliability of the results. Observer bias is a particular concern in trials where blinding was not possible due to the distinct characteristics of DACC dressings [12,32]. In these studies, knowledge of the dressing type may have influenced both clinician assessments of wound healing and patient-reported outcomes. Another source of bias is the high dropout rates or loss to follow-up in some trials [13], particularly in populations with chronic wounds or complex surgical cases. When participants drop out or are lost to follow-up, it is often unclear whether they experienced complications that could have affected the overall outcomes. These missing data can skew the results and limit the ability to draw definitive conclusions about the effectiveness of DACC dressings. The inclusion of patients with comorbidities, such as obesity or diabetes, introduces confounding variables that may not always be adequately controlled in the studies [15]. These underlying health conditions can significantly impact wound healing and infection rates, making it difficult to attribute observed outcomes solely to the use of DACC dressings.

### 6.6. Limitations of DACC Dressings

While the evidence supporting the use of DACC dressings is promising, there are several inherent limitations associated with their use that must be considered in both clinical practice and future research. One key limitation is the cost of DACC dressings, which can be higher than that of standard dressings. While preliminary studies suggest that the long-term cost savings associated with reduced infection rates may offset the initial higher cost, the upfront expense of DACC dressings may still pose a barrier to their widespread adoption, particularly in low-resource healthcare settings [80]. The cost issue is further compounded by the limited availability of DACC dressings in some regions, where hospitals and clinics may be more accustomed to using more readily available antimicrobial dressings, such as those containing silver or iodine. Another limitation is the lack of large-scale, high-quality RCTs that establish the long-term clinical and cost-effectiveness of DACC dressings across diverse patient populations and wound types. Although the existing studies provide valuable preliminary data, larger, well-designed trials are needed to confirm the benefits of DACC dressings and inform clinical guidelines. Future research should also address the limitations inherent in current study designs. For example, studies should aim to reduce biases through improved blinding methods and by ensuring that follow-up periods are sufficiently long to capture long-term outcomes, particularly in chronic wound populations. Additionally, future studies should focus on including larger, more diverse patient cohorts to ensure that the findings are generalizable across different clinical settings.

## 7. Limitations and Future Directions

This review has several limitations that should be considered when interpreting the findings. The heterogeneity of the included studies, in terms of study designs, patient populations, surgical procedures, and wound types, presents a challenge for drawing consistent conclusions. The variability in outcomes measured—such as infection rates, wound healing times, and bacterial load—complicates comparisons and prevents the formulation of a uniform evidence base for the efficacy of DACC dressings. Many of the available studies suffer from methodological weaknesses, including small sample sizes, non-randomized designs, and short follow-up durations. These limitations reduce the generalizability of the findings. Without long-term data, particularly for chronic wounds or recurrent infections, it is difficult to assess the sustained impact of DACC dressings on infection prevention and wound healing. The lack of blinding in many studies also introduces the risk of observer bias, especially in trials comparing DACC dressings to standard dressings. Furthermore, many studies have been conducted in specific surgical populations, such as cesarean section patients, with limited exploration in other high-risk groups, such as those undergoing orthopedic or oncologic surgery. Cost-effectiveness data on DACC dressings are still preliminary. While DACC dressings have demonstrated the potential to reduce healthcare costs through the prevention of surgical site infections (SSIs) and decreased use of antibiotics, comprehensive long-term economic evaluations are lacking. Studies are focused mainly on short-term financial benefits, without addressing broader healthcare system implications, such as quality-of-life improvements or reductions in antimicrobial resistance (AMR) over time.

Future research should focus on addressing the gaps in the current literature to solidify the understanding of DACC-impregnated dressings and their role in surgical wound management. Well-powered randomized controlled trials (RCTs) are essential to validate the efficacy of DACC dressings across a broader spectrum of surgical populations. Expanding research into high-risk groups, such as those undergoing orthopedic or gastrointestinal surgeries, or immunocompromised patients, would help evaluate the full potential of DACC dressings in diverse clinical settings. Moreover, future studies should ensure adequate follow-up periods to evaluate long-term outcomes, such as infection recurrence and wound healing sustainability. Studies exploring the mechanisms by which DACC dressings interact with biofilms and resistant bacterial strains are necessary. These studies should be aimed at understanding whether DACC dressings can effectively prevent the formation of biofilms, particularly in high-risk surgical wounds or in the presence of prosthetic implants, which are prone to biofilm-associated infections. Additionally, tracking microbial resistance profiles in wounds treated with DACC dressings over time could provide direct evidence of their role in mitigating AMR. Future research should prioritize comprehensive cost-effectiveness analyses, which consider both short-term and long-term financial impacts. This includes evaluating the overall cost savings associated with the prevention of complications, fewer wound dressings, reduced hospital readmissions, and minimized antibiotic use. Studies conducted in low-resource settings are particularly important for determining the global applicability of DACC dressings, especially given the economic constraints in such environments.

## 8. Conclusions

DACC-impregnated dressings represent a promising advancement in surgical wound management, offering a unique physical mechanism for bacterial removal that mitigates the risk of antimicrobial resistance. This review highlights the efficacy of DACC dressings in reducing SSIs and promoting wound healing, particularly in high-risk surgeries and chronic wounds. DACC dressings outperform traditional antimicrobial options such as silver or iodine by avoiding cytotoxicity and resistance concerns. While preliminary data suggest positive outcomes in terms of clinical efficacy and cost-effectiveness, limitations such as small sample sizes, short follow-up durations, and lack of high-quality randomized controlled trials must be addressed. Future research should focus on expanding the evidence base across diverse surgical populations, with an emphasis on long-term outcomes, antimicrobial resistance mitigation, and cost-effectiveness. With further validation, DACC dressings have the potential to become a critical tool in sustainable wound care practices globally.

## Figures and Tables

**Figure 1 ebj-06-00001-f001:**
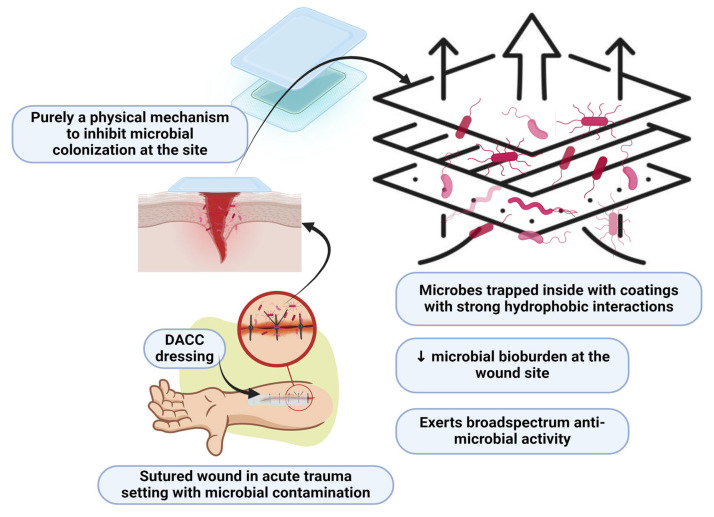
Mechanism of action of DACC dressings (created with Biorender.com on 1 September 2024).

**Figure 2 ebj-06-00001-f002:**
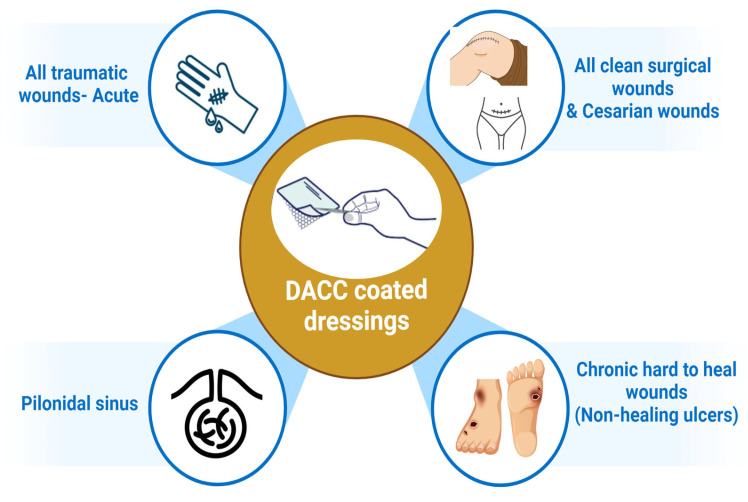
Application of DACC-coated dressing (created with Biorender.com on 1 September 2024).

**Figure 3 ebj-06-00001-f003:**
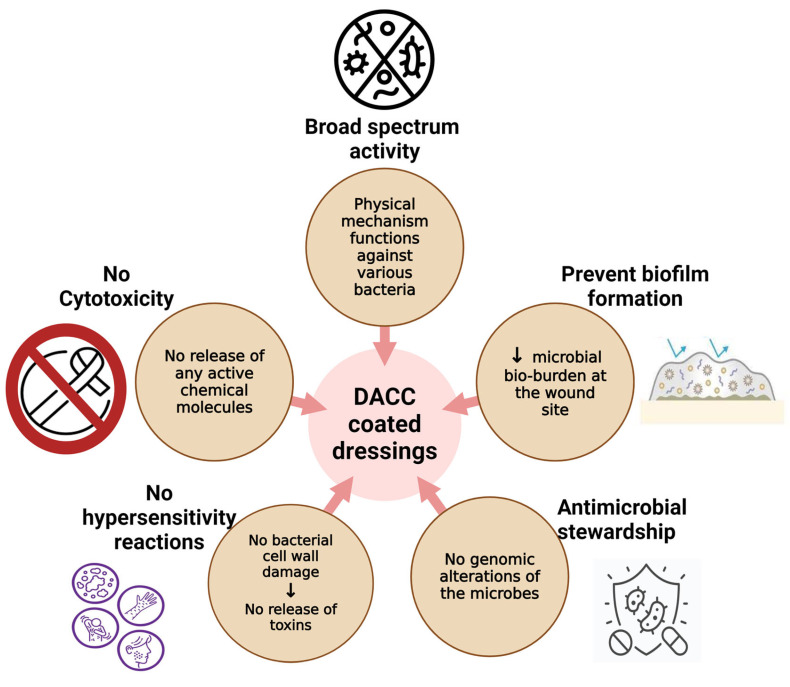
Advantages of DACC-coated dressings (created with Biorender.com on 1 September 2024).

**Table 1 ebj-06-00001-t001:** Summary of findings from the studies included in the review.

Studies	Methods	Participants	Interventions	Outcomes	Primary Findings
Meberg et al. [19] (1990)	Randomized control trial	2441 newborn infants	Alternately allocated to umbilical cord stump dressing with DACC-coated dressing or daily cleansing with chlorhexidine	Newborn infection (conjunctivitis, pyoderma, paronychia, omphalitis)	No significant difference in overall infection rates or omphalitis
Hampton et al. [20] (2007)	Case series	21 patients with non-healing wounds over 3 months old	Treated with Cutimed^®^ Sorbact^®^ as part of the treatment plan	Inflammation, exudate, malodor, wound size, pain	60% of wounds healed; 100% had reduced exudate levels; 58% had reduced odor
Kammerlander et al. [21] (2008)	Non-randomized multi-center evaluation	116 patients (62 male) treated in four European hospitals	Patients treated with Cutimed^®^ Sorbact^®^ as part of their therapeutic regime	Assessment of inflammation reduction; infection control	81% of wounds successfully treated for infection; 21% of wounds did not respond to treatment
Pirie et al. [22] (2009)	Case series	3 patients with chronic non-healing wounds	DACC-coated dressing used as the primary wound contact layer along with other therapies	Wound healing, infection evidence, wound size, exudate levels	All patients showed clinical improvement (reduced wound size and slough)
Powell et al. [23] (2009)	Case series	6 patients with various clinically infected or delayed-healing wounds	Cutimed^®^ Sorbact^®^ used as wound contact layer for 2–8 weeks	Inflammation, exudate, odor, wound healing	All wounds reduced in size, exudate, and odor; 80% completely healed
Skinner et al. [15] (2010)	Case series	4 patients with diabetic foot ulcers	Treated with Cutimed^®^ Sorbact^®^ as part of their treatment plan	Bacterial colonization, infection, wound healing	One wound completely healed; 3 progressed towards healing
Derbyshire et al. [24] (2010)	Case series	3 patients with chronic wounds over 4 years old	Treated with Cutimed^®^ Sorbact^®^ as part of their treatment plan	Wound size, healing, resource use, pain, exudate levels	Wounds became cleaner, dryer, and required fewer dressing changes
Haycocks et al. [25] (2011)	Case series	19 patients with diabetic foot ulcers, up to age 80	Treated with DACC-coated dressing as a wound contact layer for 4 weeks	Infection, healing, patient and clinician assessment	All wounds showed reduced infection signs; 69% reduced in size; 27.6% healed completely
Sibbald et al. [26] (2012)	Case series	14 patients with lower limb ulcers (diabetic foot or venous leg ulcers)	Ulcers dressed 3 times a week for 4 weeks with DACC-coated dressing	Superficial infection, total ulcer surface area, pain	Total surface area reduced from 1.74 cm^2^ to 1.15 cm^2^ (*p* = 0.337); no significant difference in infection rates
Bruce et al. [27] (2012)	Multi-center evaluation	13 patients with chronic infected wounds	Treated with DACC-coated dressings for 28 days or until infection signs resolved	Erythema, pain, heat, edema, odor, exudate	86% infection reduction; 79% wounds reduced in size
Bullough et al. [28] (2012)	Case series	4 patients with complex open abdominal wounds	DACC-coated dressings and swabs used as a wound contact layer throughout treatment	Wound infection recurrence; wound dimensions; wound healing; pain during dressing changes; exudate and odor	3 out of 4 wounds healed, and signs of infection resolved by day 14
Gentili et al. [29] (2012)	Non-comparative, double-blind, pilot study	19 patients with chronic lower limb ulcers	Wounds treated with saline rinse, surgical debridement, and DACC dressing for 4 weeks	Wound condition, quality of life, bacterial load	66% of wounds reduced in size; bacterial load decreased in all cases.
Kleintjes et al. [30] (2015)	Prospective pilot study	13 patients over 16 with burn wounds	Burns dressed with DACC-coated dressings, Cuticcot^®^, and Silverlon^®^	Wound swab MC&S, visual inspection of wounds	DACC-coated areas appeared cleaner and had less bacterial growth
Choi et al. [31] (2015)	Case series	7 patients (4 male) requiring skin grafts on clean surgical wounds	Skin grafts dressed with DACC-coated dressing and tie-over dressing for 5 days	Wounds checked for infection at 5, 14, and 30 days post-procedure	No infections were noted in the wounds
Mosti et al. [17] (2015)	Randomized, comparative, single-center study	40 patients over 18 with infected vascular ulcers over 6 months old	Randomized to silver hydrofiber dressing or DACC-coated dressing	Ulcer bacterial load	73.1% bacterial load reduction in DACC group vs. 41.6% in silver group (*p* < 0000.1)
Stanirowski et al. [12] (2016)	Single blinded, randomized control trial	543 women over 18 undergoing planned or emergency C-section	Randomized to either DACC-coated post-op dressing or standard surgical dressing	Superficial or deep SSI within 14 days after C-section (as per CDC)	SSI rates were 1.8% in DACC group vs. 5.2% in control group (*p* = 0.04)
Stanirowski et al. [32] (2019)	Single blinded, randomized, controlled pilot study	142 women over 18 years undergoing planned or emergency C-section	Randomized to either DACC-coated post-op dressing or standard surgical dressing	Superficial or deep SSI within 14 days after C-section (as per CDC)	SSI rates were 2.8% in DACC group vs. 9.8% in control group (*p* = 0.08)
Mulpur et al. [33] (2024)	Prospective, multicentric observational study	106 patients (71 orthopaedic cases and 35 gastrointestinal casses)	DACC dressing applied immediately post-surgery and assessed over 30 days for the incidence of superficial or deep SSI	1.9% cases of SSI were reported in orthopaedic patients	73.5% patients reported an improved pain experiences during dressing changes compared to previous dressings.

**Table 2 ebj-06-00001-t002:** A summary of these dressings compared to the DACC dressings.

Feature	Silver-Coated Dressings	Iodine Dressings	Honey-Based Dressings
**Mechanism**	Releases silver ions with antimicrobial properties	Releases iodine with antiseptic properties	Contains natural antibacterial properties
**Antimicrobial action**	Broad-spectrum antibacterial, anti-inflammatory, and anti-oxidative	Broad-spectrum antibacterial, wound debridement, and odour control	Broad-spectrum antibacterial, promotes healing, and reduces inflammation
**Differences**	Uses silver nanoparticles for enhanced efficacy	Uses cadexomer iodine for controlled release	Uses natural honey for its healing properties
**Similarities**	Both are used for chronic wound management and infection control	Both are used for chronic wound management and infection control	Both are used for chronic wound management and infection control

## Data Availability

All data are available within the manuscript.

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
