# Peer review of "Efficacy of Dialkylcarbamoylchloride (DACC)-Impregnated Dressings in Surgical Wound Management: A Review"

_2673-1991, 2025, doi:10.3390/ebj6010001_

Round 1
Reviewer 1 Report
Comments and Suggestions for Authors
Dear Authors,
Thank you for the DACC review submission.
A few noted comments:
1) On lines 72-77, can you please provide citations and evidence to support these statements?
2) On figure #1 please list the date of the biorender creation.
3) In section 2.2 (lines 108-118) you list "A significant body of evidence supports" yet only describe two studies. For a review paper, I would imagine additional would need to be reported for SSIs (differences in the studies, similarities, etc.) What is novel with respect to SSIs and using DACC from the last total DACC published review?
4) In section 2.3 (line 121), I would suggest creating a table describing the 17 studies and the overall outcomes.
5) On the "Comparisons with Antimicrobial Dressings, please include an example image of the following: Silver-coated Dressings, Iodine Dressings, and Honey-Based Dressings, etc. etc etc." I would also suggest a comprehensive table of those products available and currently being clinically used, including the mechanism, differences, and similarities in the table.
6) In section 5.3 cost effectiveness is discussed, yet no cost values (even estimates) are provided anywhere in this subsection. Please find literature that includes monetary values of cost-effectiveness to enhance this section.
7) 5.5.4 Potential Biases. This section claims bias exists in the literature and the studies, yet no references or studies are given to support this. Please list areas of bias with specific citations.
8) General note: how were the studies found, what methods were used to determine the articles to be included in this review, keyword searches, what search engine, etc? For example, this article, see below details this specifically on DACC:
Experimental and clinical evidence for DACC-coated dressings: an update Mark Rippon, Alan A Rogers, Karen Ousey, and Paul Chadwick Journal of Wound Care 2023 32:Sup8a, S13-S22Overall, I see a strong need for an update to include additional and more comprehensive document searches, reviews, and inclusions in your paper. This includes how your DACC review paper is vastly different from others in the literature, including the materials and methods of your review article.
Author Response
We thank the reviewers and editorial team for making their efforts to improve the article to increase its value for publication. Herewith we submit the revised version of the article addressing the reviewer’s comments and the action taken for their valuable suggestions has been mentioned below.
Comment 1:
On lines 72-77, can you please provide citations and evidence to support these statements?
Reply: DACC is a fatty acid derivative that is highly hydrophobic. Micro-organisms commonly
responsible for causing SSI or colonising chronic wounds generally have hydrophobic extracellular surfaces, and will therefore irreversibly adhere to the DACC coating on dressings. Subsequent dressing changes will then result in the removal of large numbers of microbes and a decreased bacterial load at the wound site. Mechanical removal of bacteria comes with several additional potential advantages; DACC coated dressings have shown no evidence of wound or systemic absorption of dressing component, or adverse reactions other than to the adhesive component of the dressing. Perhaps most importantly, since the mechanism of antibacterial action is of physical binding and removal, there is no risk of bacteria developing resistance, and the lack of bacteriolysis prevents endotoxin release to
the wound bed. Leukomed® Sorbact®, an example of a DACC-coated dressings used with this principle.
- Ljungh A, Wadström T. Growth conditions influence expression of cell surface
hydrophobicity of staphylococci and other infection pathogens. Microbiol Immunol
1995;39:753-57. - Ljungh A, Yanagisawa N, Wadström T. Using the principle of hydrophobic interaction to
bind and remove wound bacteria. Journal of Wound Care 2006;15:175-80. doi:
10.12968/jowc.2006.15.4.26901 - von Hallern B, Lang F. Has Cutisorb® Sorbact® proved its practical value as an
antimicrobial dressing? Medizin & Praxis Spezial - Infected wounds, 2005 - Totty, J.P.; Bua, N.; Smith, G.E.; Harwood, A.E.; Carradice, D.; Wallace, T.; Chetter, I.C. Dialkylcarbamoyl Chloride (DACC)-Coated Dressings in the Management and Prevention of Wound Infection: A Systematic Review. J Wound Care 2017, 26, 107–114, doi:10.12968/jowc.2017.26.3.107.
- Ortega-Peña, S.; Chopin-Doroteo, M.; Tejeda-Fernández de Lara, A.; Giraldo-Gómez, D.M.; Salgado, R.M.; Krötzsch, E. Dialkyl Carbamoyl Chloride-Coated Dressing Prevents Macrophage and Fibroblast Stimulation via Control of Bacterial Growth: An In Vitro Assay. Microorganisms 2022, 10, 1825, doi:10.3390/microorganisms10091825.
- Malone, M.; Radzieta, M.; Schwarzer, S.; Walker, A.; Bradley, J.; Jensen, S.O. In Vivo Observations of Biofilm Adhering to a Dialkylcarbamoyl Chloride-Coated Mesh Dressing When Applied to Diabetes-Related Foot Ulcers: A Proof of Concept Study. Int Wound J 2023, 20, 1943–1953, doi:10.1111/iwj.14054.
Comment 2:
On figure #1 please list the date of the biorender creation.
Reply:
Date added as suggested
Comment 3:
In section 2.2 (lines 108-118) you list "A significant body of evidence supports" yet only describe two studies. For a review paper, I would imagine additional would need to be reported for SSIs (differences in the studies, similarities, etc.) What is novel with respect to SSIs and using DACC from the last total DACC published review?
Reply:
Thanks to the reviewer for the comment. We have now added additional studies to the revised manuscript to support the validity of DACC dressings in SSI reduction.
Comment 4:
In section 2.3 (line 121), I would suggest creating a table describing the 17 studies and the overall outcomes.
Reply:
Thanks to the reviewer for the comment. We have now included the table as suggested summarizing the evidence on DACC dressings in chronic wound management as suggested.
Comment 5
On the "Comparisons with Antimicrobial Dressings, please include an example image of the following: Silver-coated Dressings, Iodine Dressings, and Honey-Based Dressings, etc. etc etc." I would also suggest a comprehensive table of those products available and currently being clinically used, including the mechanism, differences, and similarities in the table.
Reply:
Thanks for the comment. We have now included a table summarizing the mechanisms, differences and similarities of the dressings described as suggested.
Comment 6
In section 5.3 cost effectiveness is discussed, yet no cost values (even estimates) are provided anywhere in this subsection. Please find literature that includes monetary values of cost-effectiveness to enhance this section.
Reply:
Thanks for the comment. We have now included the actual cost data from the studies and pointed out case examples and cost-saving principles utilized in these studies and their context in different country scenarios.
Comment 7
5.5.4 Potential Biases. This section claims bias exists in the literature and the studies, yet no references or studies are given to support this. Please list areas of bias with specific citations.
Reply
Thanks for the comment. We have not pointed the specific comments to the individual studies where the bias is noted. Thanks again for the comment to add more specificity to the content provided.
Comment 8
General note: how were the studies found, what methods were used to determine the articles to be included in this review, keyword searches, what search engine, etc? For example, this article, see below details this specifically on DACC:
Reply
Thanks for the insightful comment to add more transparency to the methodology utilized. We have now included a methodology section to elaborate on the methods utilized to shortlist the articles utilized for review process as suggested.
Comment 9
Experimental and clinical evidence for DACC-coated dressings: an update Mark Rippon, Alan A Rogers, Karen Ousey, and Paul Chadwick Journal of Wound Care 2023 32:Sup8a, S13-S22
Overall, I see a strong need for an update to include additional and more comprehensive document searches, reviews, and inclusions in your paper. This includes how your DACC review paper is vastly different from others in the literature, including the materials and methods of your review article.
Reply
The article highlighted as included only 9 studies of which only 5 were clinical and 4 were laboratory studies. In this current review, we have included 80 studies of relevance to DACC dressing and we consider this to be the most comprehensive summary of relevant literature on the DACC dressings to date.
Reviewer 2 Report
Comments and Suggestions for Authors
Dear authors,
The manuscript is detailed and well written. I do agree that there is limited awareness regarding DACC dressings and its clinical applications. You have documented the benefits as well as the gaps in literature very well.
However, there are certain minor observations
1. Abstract
Replace the word synthesizes in the statement “This review synthesizes current evidence...”
2. Line 88- The authors have mentioned “DACC-coated dressings have emerged as a promising tool in wound management, primarily due to their unique mechanism of action, which binds and removes bacteria from the wound bed without releasing antimicrobial agents”.
If the antimicrobial agents are not released, what is the purpose of antimicrobial dressings? Does the statement mean that antimicrobial agents need not be incorporated?
3. Line 101- “Unlike traditional antimicrobial dressings, DACC-coated dressings do not rely on chemical agents to kill bacteria, which reduces the risk of developing antimicrobial resistance”
This being mention, why have the authors mentioned “exerts broad spectrum anti-microbial activity” in Figure 1?
4. Are DACC dressings capable of managing wound exudate?
5. Line 461- “One key limitation is the cost of DACC dressings, which can be higher than that of standard dressings”
Which are the other standard dressings? Since the authors have mentioned that DACC dressings are cheaper than silver impregnated dressings.
Author Response
Comment 1: The manuscript is detailed and well written. I do agree that there is limited awareness regarding DACC dressings and its clinical applications. You have documented the benefits as well as the gaps in literature very well.
Reply: We thank the reviewers and editorial team for taking their efforts to improve the article to increase its value for publication. Herewith we submit the revised version of the article addressing the reviewer’s comments and the action taken for their valuable suggestions has been mentioned below.
Comment 2: Replace the word synthesizes in the statement “This review synthesizes current evidence...”
Reply: Revised as suggested in the abstract
Comment 3:
Line 88- The authors have mentioned “DACC-coated dressings have emerged as a promising tool in wound management, primarily due to their unique mechanism of action, which binds and removes bacteria from the wound bed without releasing antimicrobial agents”.
If the antimicrobial agents are not released, what is the purpose of antimicrobial dressings? Does the statement mean that antimicrobial agents need not be incorporated?
Reply: Thanks for the comment. We would like to explain more on the mechanism of action of DACC dressings to clarify this. DACC is a fatty acid derivative that is highly hydrophobic. Microorganisms commonly responsible for causing SSI or colonising chronic wounds generally have hydrophobic extracellular surfaces, and will therefore irreversibly adhere to the DACC coating on dressings. Subsequent dressing changes will then result in the removal of large numbers of microbes and a decreased bacterial load at the wound site. Mechanical removal of bacteria comes with several additional potential advantages; DACC-coated dressings have shown no evidence of wound or systemic absorption of dressing component, or adverse reactions other than to the adhesive component of the dressing. Perhaps most importantly, since the mechanism of antibacterial action is of physical binding and removal, there is no risk of bacteria developing resistance, and the lack of bacteriolysis prevents endotoxin release to the wound bed. Leukomed® Sorbact®, an example of a DACC-coated dressings. We hope we clarified the concept of action of DACC dressings.
Comment 4:
“Unlike traditional antimicrobial dressings, DACC-coated dressings do not rely on chemical agents to kill bacteria, which reduces the risk of developing antimicrobial resistance”
This being mention, why have the authors mentioned “exerts broad spectrum anti-microbial activity” in Figure 1?
Reply: Thanks to the reviewer for the comment. As explained earlier, the extent of anti-microbial activity is mentioned broad because the physical property of adhesion is applicable across a broad range of microbial organisms with hydrophobic outer layer of adhesion. We hope this explain the broad spectrum mentioned in Figure 1.
Comment 5:
Are DACC dressings capable of managing wound exudate?
Reply: DACC dressings have been shown to significantly reduce wound odour, exudate and size compared to normal dressings. The below studies have validated the same.
Bullough L, Little G, Hodson J, et al. The use of DACC-coated dressings for the treatment
of infected, complex abdominal wounds. Wounds UK 2012;8:102-09.
Hampton S. An evaluation of the efficacy of Cutimed® Sorbact® in different types of nonhealing wounds. Wounds UK 2007;3:113-19.
Powell G. Evaluating Cutimed Sorbact: using a case study approach. The British journal of
nursing 2009;18:S30, S32-S36.
Pirie G, Duguid K, Timmons J. Cutimed® Sorbact® gel: A new infection management
dressing. Wounds UK 2009;5:74-78
Comment 6:
Line 461- “One key limitation is the cost of DACC dressings, which can be higher than that of standard dressings”
Which are the other standard dressings? Since the authors have mentioned that DACC dressings are cheaper than silver impregnated dressings.
Reply: Thanks to the reviewer for the comment. By standard dressings, we mean the regular non-coated dressings. However, compared to the silver-impregnated dressings DACC dressings are cheaper. Hence the statement. Hope we made it clear now.
Round 2
Reviewer 1 Report
Comments and Suggestions for Authors
Dear Authors,
Thank you for the comprehensive updates. This reads much better and I agree with publication in its present form.
Best.
Reviewer 2 Report
Comments and Suggestions for Authors
Dear authors,
Thank you for providing your responses.